# Therapeutic Strategies for Spinocerebellar Ataxia Type 1

**DOI:** 10.3390/biom13050788

**Published:** 2023-05-02

**Authors:** Laurie M. C. Kerkhof, Bart P. C. van de Warrenburg, Willeke M. C. van Roon-Mom, Ronald A. M. Buijsen

**Affiliations:** 1Department of Human Genetics, Leiden University Medical Center, 2333 ZA Leiden, The Netherlands; 2Dutch Center for RNA Therapeutics, Leiden University Medical Center, 2333 ZA Leiden, The Netherlands; 3Department of Neurology, Donders Institute for Brain, Cognition, and Behaviour, Radboud University Medical Center, 6525 GA Nijmegen, The Netherlands

**Keywords:** Spinocerebellar ataxia type 1, neurodegenerative disorder, therapeutic strategies, preclinical research, clinical trials, gain-of-function mechanism, ataxin1, *ATXN1*, polyQ disorders

## Abstract

Spinocerebellar ataxia type 1 (SCA1) is an autosomal dominant neurodegenerative disorder that affects one or two individuals per 100,000. The disease is caused by an extended CAG repeat in exon 8 of the *ATXN1* gene and is characterized mostly by a profound loss of cerebellar Purkinje cells, leading to disturbances in coordination, balance, and gait. At present, no curative treatment is available for SCA1. However, increasing knowledge on the cellular and molecular mechanisms of SCA1 has led the way towards several therapeutic strategies that can potentially slow disease progression. SCA1 therapeutics can be classified as genetic, pharmacological, and cell replacement therapies. These different therapeutic strategies target either the (mutant) *ATXN1* RNA or the ataxin-1 protein, pathways that play an important role in downstream SCA1 disease mechanisms or which help restore cells that are lost due to SCA1 pathology. In this review, we will provide a summary of the different therapeutic strategies that are currently being investigated for SCA1.

## 1. Introduction

Spinocerebellar ataxia type 1 (SCA1) is a progressive, autosomal dominant neurodegenerative disorder characterized by ataxia, speech, and swallowing difficulties, spasticity, and abnormal control of eye movements. At later stages of the disease, patients show signs of muscular atrophy and cognitive defects [1,2,3,4]. Eventually, patients show bulbar dysfunction that leads to respiratory failure, which is the main cause of death [5]. In SCA1, the first clinical signs may develop between 4 and 74 years of age, but they typically manifest in the third or fourth decade of life [6]. The time span from onset of the disease to death varies from 10 to 30 years, with an average of 15 years. Individuals with a more juvenile onset show a more rapid disease progression [7]. Epidemiological information about the prevalence of SCA1 is limited to only a few studies but is thought to be one to two cases per 100,000 people [6,8,9,10]. 

SCA1 is caused by an expanded CAG repeat in the coding region of the human *ATXN1* gene [2]. Normal alleles range from 6 to 38 CAG repeats, with 1 to 3 CAT interruptions that are thought to be involved in the stability of the trinucleotide stretch during DNA replication. Disease-causing alleles have 39 to 44 CAG repeats without stabilizing CAT interruptions or larger expansions with CAT interruptions [11,12,13,14]. There is an inverse correlation between the number of uninterrupted CAG repeats and the age of disease onset [13,15]. The *ATXN1* gene is organized in nine exons of which the first seven fall in the 5′ untranslated region (UTR). The 5′UTR exons can undergo alternative splicing, suggesting that transcriptional and translational regulation of the SCA1 encoded protein, ataxin-1, may be complex [16,17]. The protein is expressed throughout the brain and is present mainly in the nucleus of neuronal cells, where it is thought to play a role in transcriptional regulation. However, in Purkinje Cells (PC) in the cerebellum, the ataxin-1 protein is localized in both the cytoplasm and the nucleus [16]. Expansion of the *ATXN1* gene results in an expanded polyglutamine tract in the ataxin-1 protein, thereby inducing a polyglutamine-induced, toxic gain-of-function mechanism leading to SCA1 disease pathology [18].

Currently, there is no disease-modifying treatment available for SCA1 patients, but several therapies, including physiotherapy, occupational therapy, and speech therapy are offered which provide some symptomatic relief and help in improving the quality of life [19]. Multiple disease-modifying treatments are currently being developed. In this review, we will discuss SCA1 pathophysiology, followed by an outline of the therapeutic strategies that are being investigated, limiting our review to studies that have been tested in animal models of SCA1 or have already progressed to clinical trials. We have divided the strategies into the following three categories: (1) modulation of ataxin-1 levels, (2) pharmacological targets, and (3) cell replacement therapies. Lastly, we discuss some of the limitations that hamper drug development for SCA1 and outline our future perspectives on SCA1 drug development.

## 2. SCA1 Pathophysiology

SCA1 pathophysiology is characterized mainly by degeneration of the cerebellar cortex, deep cerebellar nuclei, and brainstem [3]. Loss of PCs from the cerebellar cortex and the inferior olivary nucleus is most prominent, but a loss of neurons in cortical, subcortical, and spinal structures is also observed [20]. SCA1 patients also show degeneration of the peripheral nervous system (PNS), with a loss of motor neurons in the lumbosacral spinal cord and the thin anterior spinal roots [21]. Additionally, a loss of myelin as well as an early activation of astrocytes and microglia have been observed [22,23]. 

In the last decades, research has been conducted to study the molecular mechanisms behind SCA1 pathogenesis. However, the disease mechanisms are still not fully understood [24,25]. Genetic studies have shown that *Sca1* knock-out mice do not display any signs of ataxia, motor incoordination, or cerebellar degeneration [26]. However, *Sca1* knock-out mice do demonstrate deficits in several learning and memory behavioral tests and neurophysiological studies. These results show that SCA1 is not caused by a loss-of-function mechanism and indicate a role for ataxin-1 in learning and memory [26]. On the other hand, expression of mutant ataxin-1 in mouse and *Drosophila* models leads to relevant disease phenotypes and pathology, suggesting that SCA1 is caused by a toxic gain-of-function mechanism [27,28,29]. 

The polyglutamine repeat expansion is thought to exert its toxicity by altering the interactions of several factors with key domains, including the C-terminus and AXH domain in ataxin-1 (Figure 1, [30,31]). The ataxin-1 C-terminus contains a highly conserved region, with a nuclear localization signal (NLS) and domains for binding of the 14-3-3 protein, which is a multifunctional regulatory molecule, as well as RBM17, which is a transcription factor (Figure 1, [30,31,32]). The NLS shuttles the ataxin-1 protein from the cytoplasm to the nucleus. Mice expressing mutant ataxin-1 without NLS did not show any SCA1 disease signatures, while mice expressing mutant ataxin-1 containing a deletion within the self-association region spanning amino acids 495–605 developed ataxia and PC pathology in the absence of nuclear ataxin-1 aggregates [33,34,35]. This demonstrates that nuclear ataxin-1 localization is critical to develop pathology [30]. Other research suggests that aggregation is important in SCA1 pathology, as several genetic modifiers identified in a *Drosophila* screen highlight the role of protein folding and clearance in SCA1 [29]. Furthermore, it is hypothesized that phosphorylation of ataxin-1 at S776 facilitates 14-3-3 protein binding, thereby stabilizing the ataxin-1 protein, causing accumulation and subsequent neurodegeneration in SCA1 [31,36,37,38]. The expanded CAG repeat in ataxin-1 has also been shown to enhance the formation of the protein-complex-containing transcription factor RBM17 [39]. 

Ataxin-1 additionally contains an AXH domain of 120 amino acids, which mediates many protein–protein interactions and RNA-binding activity [24,25]. The AXH domain forms a complex with the transcriptional repressor capicua (CIC), and mutant ataxin-1 reduces complex formation and repressor function of CIC in both *Drosophila* and cell models. This contributes to SCA1 pathogenesis through a partial loss-of-function mechanism [39,40]. By inhibiting this repressor activity, gene expression of glutamatergic receptor genes and synaptic long-term depression in the PCs is altered, which contributes to disease pathophysiology [40]. Indeed, alterations in PC firing output are observed in SCA1, which are thought to be caused by alterations in the PC synaptic input as well as changes in the intrinsic PC firing [41]. Mutations in the binding site of CIC with the AXH domain in SCA1^154Q/2Q^ mice normalized genome-wide CIC binding. However, transcriptional and behavioral phenotypes were only partially rescued, suggesting the involvement of additional factors in SCA1 disease pathogenesis [34,42]. The AXH domain is also known to bind to several other transcriptional regulators, including RORα and Tip60, the RoRα co-activator, thereby mediating the expression of a group of genes that play a role in PC development and function [43,44]. Additionally, AXH acts as a dimerization domain for ataxin-1-like (*ATXN1L). ATXN1L*, or Boat (brother-of-ataxin-1), shares 33% homology with *ATXN1*, including the conserved AXH domain, where both *ATXN1* and *ATXN1L* interact with the transcriptional repressor CIC [24,25,45]. The AXH domain also binds to RNA, and binding is dependent on the size of the CAG repeat expansion in such a way that the ability of *ATXN1* binding to RNA decreases when the repeat expansion increases, possibly altering its role in RNA metabolism [46]. 

## 3. Therapeutic Strategies

SCA1 therapeutics can be tested in several disease model systems, including cell culture and animal models. Here, we provide a short overview of the most important mouse models used for the development of SCA1 therapeutics. Subsequently, a summary of treatment strategies targeting ataxin-1 levels, SCA1 relevant pathways, or cell replacement studies for SCA1, which have been tested in preclinical animal models or are in clinical studies, will be discussed. SCA1 is modeled in different animals, including fly, zebrafish, and mouse models [27,28,29,47]. The most frequently used SCA1 mouse models are the SCA1 transgenic (B05) and the knock-in SCA1 mouse model, SCA1^154Q/2Q^ (Table 1, [27,28]).

The B05 transgenic mouse model uses the Purkinje cell protein 2 (Pcp2) promoter to ensure overexpression of human *ATXN1* cDNA in the cerebellar PCs specifically [27]. The human transgene was modified to contain 82 uninterrupted CAG repeats. Overexpression of this transgene in mice resulted in an ataxic phenotype as well as neuropathological changes [27,48]. Mice showed a significant loss of PCs in the cerebellum and accumulation of nuclear inclusions in the PCs [27,48]. However, the transgenic mouse model does not reflect the full disease pathology, which also involves a dysfunction in a variety of other neurons next to PCs, leading to, for instance, the cognitive dysfunction observed in humans [3,49].

The SCA1^154Q/2Q^ knock-in mice expresses mutant *Atxn1,* with 154 CAG repeats under the control of endogenous regulators, thereby expressing mutant *Atxn1* throughout the brain and spinal cord. SCA1^154Q/2Q^ mice display motor incoordination and muscle wasting, as well as premature lethality, reduced adult hippocampal neurogenesis, and kyphosis (exaggerated, forward rounding of the upper back due to weakness in the spinal bones) [28]. Furthermore, the mice display cognitive deficits, such as reduced memory and cognitive flexibility [50,51].

In addition to mouse models to study SCA1, a fly model was developed [29,47]. The *Drosophila melanogaster* fly model expresses human 82Q *ATXN1* in the eye retina using the GAL4/UAS system. This leads to nuclear inclusion formation in the eye photoreceptor cells and neurons of the CNS, subsequently resulting in degeneration of the retina and neurodegeneration [29]. However, the fly model lacks a cerebellum, making it difficult to model the cerebellar neurodegeneration of SCA1 [47]. 

### 3.1. Modulating Ataxin-1 Levels

As discussed before, the cellular pathological mechanisms underlying SCA1 are driven mainly by the presence of the mutant ataxin-1 protein. Presently, many intervention strategies focus on the development of therapies that can reduce the levels of mutant ataxin-1, which will have an effect on all downstream pathological processes. The ataxin-1-lowering strategies that are currently in development target *ATXN1* RNA act by either modulating RNA levels or by blocking translation. Here, we will provide an overview of the different ataxin-1 modulating therapies that are currently being explored for SCA1 (Figure 2).

#### 3.1.1. RNA Interference

RNA interference (RNAi), or post-transcriptional gene silencing, is a conserved biological process in which noncoding double-stranded RNA molecules are involved in suppression of gene expression through translational or transcriptional repression. [52,53]. The process can be induced by tailor-made genetic sequences, thus providing a method for deliberately silencing a gene of interest in model systems [53]. The RNAi process is mediated by three functionally different noncoding dsRNA molecules, namely microRNA (miRNA), short hairpin RNA (shRNA), and small interfering RNA (siRNA), which converge into the same RNAi pathway. In general, RNAi strategies cleave the ds-RNA with an endonuclease called Dicer into single RNA fragments. The guide strands of these RNA fragments are then subsequently loaded into an RNA-induced silencing complex (RISC) to promote endonucleolytic cleavage of the homologous mRNA by the RNase Argonaute 2 (Figure 2, [54]). 

A first proof-of-concept study for SCA1 in 2004 showed the ability of RNAi to inhibit neurodegeneration in the SCA1 B05 transgenic mouse model. Upon intracerebellar injection, recombinant adeno-associated virus (AAV) vectors expressing shRNAs targeting human *ATXN1* sequences improved motor coordination, restored cerebellar morphology, and resolved ataxin-1 inclusions in PCs of these mice (Table 2, [55]). RNAi therapies can be delivered using lentiviruses and AAVs, as they are minimally immunogenic and result in the stable expression in the brain regions of interest [56]. Keiser, Davidson, and colleagues delivered improved viral vectors expressing miRNAs targeting human ataxin-1 (miS1 vectors) to the cerebellum of pre-symptomatic SCA1 B05 transgenic mice. This resulted in widespread cerebellar PC transduction and improved behavioral and histological phenotypes [57]. Next, to assess the extra-cerebellar therapeutic effects, the miS1 vector was injected into the deep cerebellar nuclei of pre-symptomatic SCA1^154Q/2Q^ knock-in mice, where the mutant ataxin-1 is expressed throughout the brain. This approach reduced ataxin-1 expression in the cerebellar cortex and brainstem. Cerebellar lobule integrity, rotarod performance, and histological phenotypes were preserved, and disease-related transcriptional changes were prevented for over a year, suggesting that delivery to deep cerebellar nuclei is sufficient for widespread therapeutic benefit [58]. Importantly, as treatment in humans is started after symptom onset, a follow-up study in symptomatic transgenic B05 mice was performed. Results of this study showed that the SCA1 phenotypes could be reversed when the miS1 vector was delivered after symptom onset [59]. To study biodistribution and tolerability, the compound was subsequently administered into the deep cerebellar nuclei of adult rhesus macaques. Eight weeks after injection, transduction was seen in SCA1 relevant brain regions, including the deep cerebellar nuclei, cerebellar PCs, and the brainstem, where a significant reduction of endogenous *ATXN1* mRNA levels of more than 30% compared with the uninjected hemisphere was found [60]. There were no clinical complications as assessed by behavioral abnormalities or neuropathological findings [60]. Altogether, these data are supportive of a clinical application of an AAV-based RNAi therapy for SCA1 [57,58,59,60]. To advance this technology to patients, investigational new drug (IND)-enabling studies were performed. However, 3 months after AAV-based delivery into the deep cerebellar nuclei of rhesus macaques, cerebellar toxicity was observed [61]. RNA-sequencing studies showed that, despite limited amounts of vector, there was substantial 3′ inverted terminal repeat promoter activity showing neurotoxic effects in the rhesus macaques [61]. By altering the miS1 expression context, the promoter activity was reduced. These findings stress the importance of extended safety studies in multiple species when assessing new therapeutics for human application [61].

#### 3.1.2. Antisense Oligonucleotides

A second RNA-based therapy approach to reduce gene or protein expression is the use of antisense oligonucleotides (ASOs). ASOs are chemically modified, single-stranded oligonucleotides of generally 12 to 30 bases, which bind to RNA through Watson–Crick–Franklin base pairing. Depending on the sequence and modifications, ASOs can alter RNA functions through several mechanisms. They can be used to reduce expression of a toxic protein, to modify mutant proteins, to reduce their toxicity, or to restore protein expression (Figure 2, [66]). 

ASO therapies can be delivered through intravenous administration or directly into the CSF via intrathecal (IT) or intracerebroventricular (ICV) injections [54,56,67]. Although systemic administration is less invasive and, therefore, favorable for clinical applications, it requires doses that are approximately 100 times higher compared with ICV, which increases the risk of toxicity [68]. To minimize toxicity, transporters or drug carriers can be used, which allow the ASOs to cross the blood–brain barrier (BBB) [68,69,70,71,72]. To avoid the BBB, ASOs can also be delivered directly in the CSF through IT or ICV injections [54,56,67,68]. Local delivery into the CSF is preferred, as this allows for direct and local targeting, thereby reducing the required doses and reducing systemic exposure and toxicity as well as renal and hepatic elimination [67,68]. It has been shown that after IT and ICV injections, ASOs are widely distributed in the brain, although the delivery into deeper cerebellar brain structures seems to be less efficient [68]. IT injections have already been successfully used in human clinical trials for amyotrophic lateral sclerosis and spinal muscular atrophy, without major side effects [73,74]. However, as these injections are quite invasive and treatment with ASOs are transient, re-administration of the ASOs every 3 to 4 months limits its clinical use [66,67,68].

Gapmers are chimeric ASOs that contain a central block of DNA nucleotides flanked by modified RNA sequences, usually containing 2′-O-modified chemistries. The modified sequences improve target affinity and stability, while the central DNA sequence forms a DNA/RNA hybrid, which stimulates RNA cleavage through the recruitment of RNase H (Figure 2). Following a single ICV injection of an *Atxn1* gapmer ASO353 in pre-symptomatic SCA1^154Q/2Q^ mice, a reduction of *Atxn1* levels at 6 and 24 weeks after ASO administration was shown. Furthermore, a rescue of disease-associated phenotypes was demonstrated, including motor performance, survival, analysis of neurochemicals, and transcriptional disease signatures (Table 2, [62]). 

Steric or RNA blocking ASOs can inhibit (or activate) protein translation through binding to regulatory elements, including the upstream open reading frame (Figure 2, [66]). Using a 2′-O-methyl-modified (CUG)^7^ ASO with a phosphorothioate backbone that specifically targets expanded CAG stretches and does not activate RNase H-dependent RNA degradation, a lowering of polyglutamine protein levels was seen in polyglutamine patient cell lines and mouse models [64,75]. By administering 75 or 150 μg of (CUG)^7^ by ICV infusion into the right lateral ventricle of SCA1^154Q/2Q^ mice weekly for a total of 8 weeks, a reduction of *Atxn1* levels was seen in all SCA1-relevant brain regions, including cerebellum, brainstem, and spinal cord. Quantification of ASO levels showed a widespread distribution of (CUG)^7^ in all brain regions. Most importantly, ASO levels in the different brain regions and the spinal cord were similar, despite the differences in distance from the injection site [64].

In addition to delivery of the ASO to the brain, targeting *ATXN1* using RNA therapy faces another challenge, as a complete loss of *ATXN1* may contribute to Alzheimer’s disease by increasing the transcription of BACE1, which encodes for the ß-secretase enzyme [25,76]. In addition to these possible adverse effects after a loss of ataxin-1, another study investigated the change in BACE1 levels after downregulating ataxin-1 with the gapmer ASO353 [63]. Results from this study showed that ASO-mediated reduction of *Atxn1* did not lead to an unwanted increase in BACE1 levels in SCA1^154Q/2Q^ mice after 3 ASO353 ICV injections [63]. This suggests that a partial loss of ataxin-1 function later in life due to an ASO or other RNA therapy is safe and can be used in clinical applications [63]. However, to preserve the expression and function of ataxin-1, specific silencing of the expanded CAG repeat allele can also be achieved. This can be accomplished by either targeting single nucleotide polymorphisms (SNPs) or by specifically restoring the unexpanded protein [77,78,79]. 

#### 3.1.3. *ATXN1L* Overexpression

A different therapeutic strategy to treat SCA1 is overexpression of the *ATXN1* paralog, *ATXN1L* (Figure 2). Gene duplication of *Atxn1l* in SCA1^154Q/2Q^ mice reduced neuropathology and behavioral deficits through displacement of mutant ataxin-1 from its native complex with CIC, an interaction that is involved in SCA1 cerebellar pathology [45]. Two studies have shown that treatment of AAVs expressing human *ATXN1L* improves motor coordination and pathology in pre-symptomatic SCA1 B05 mice [57,65]. Given the fact that the toxic gain-of-function mechanism can be treated with the previously described RNAi or ASO-mediated knockdown of ataxin-1, overexpression of *ATXN1L* might help by also treating the loss-of-function caused by these ataxin-1 lowering therapies. When combining upregulation of *ATXN1L* with RNAi-induced downregulation of *ATXN1*, the improvement of gene expression changes and motor behavior impairments for the combined treatment were greater than for the individual treatments [65]. However, as overexpression of wildtype ataxin-1 (30Q) in *Drosophila* results in neurodegenerative phenotypes similar to those caused by the expanded protein, it is essential to monitor possible side effects of increased *ATXN1L* [29].

#### 3.1.4. S776 Phosphorylation

Since phosphorylation of ataxin-1 at the serine 776 residue (*ATXN1*-pS776) was shown to play a significant role in protein toxicity (Figure 2, [38], inhibition of S776 phosphorylation could be a promising therapeutic target for SCA1. In SCA1 cell models, pharmacological inhibition of S776 phosphorylation led to a decrease in ataxin-1 [38]. Depending on the brain region, phosphorylation of ataxin-1 at S776 is regulated by different factors. In the brainstem, phosphorylation is modulated mainly by Rsk3, whereas in the cerebellum, it is modulated via Msk1 [80]. Furthermore, protein kinase A (PKA) can mediate phosphorylation of S776. B05 mice with a mutation in PKA, generated via a Cre-lox system, showed a reduced PKA-mediated phosphorylation of *ATXN1*-S776 in PCs as well as enhanced degradation of ataxin-1 and improved cerebellar-dependent motor performance [38]. In the SCA1^154Q/2Q^ mouse model, where the S776 phosphorylation site was mutated to an Ala776 (S776A) on either the wildtype allele or the mutant allele, it was shown that targeting S776 phosphorylation could ameliorate SCA1 pathology [36,81]. Only disruption of the S776 phosphorylation site on the mutant allele, but not the wildtype allele, reduced ataxin-1 protein levels and ameliorated pathological disease hallmarks, including the reduced thickness of the cerebellar molecular layer and nuclear inclusion formation. Furthermore, there was a partial rescue of motor function and an expanded life span, but learning and memory deficits were not rescued [36]. As therapeutic approaches are likely to decrease S776 phosphorylation on both alleles, these studies show the importance of developing allele-specific intervention strategies for SCA1. Furthermore, the region-specific modulation of ataxin-1 S776 phosphorylation, by Rsk3 and Msk1 in the brainstem and cerebellum, respectively, underlines the need for combinatorial treatment targeting both kinases, which could improve therapeutic efficacy for SCA1 [80].

### 3.2. Pharmacological Targets

There are several mediators that are known to play a role in SCA1 pathology that could be potential pharmacological targets. For instance, activity of the PCs can be influenced by several drugs that target synaptic and non-synaptic receptors (Figure 3, [82,83,84]). PCs receive excitatory synaptic input from climbing fibers (CF) and parallel fibers (PF), whereas it receives inhibitory inputs from the molecular interneurons, the basket cells and stellate cells (Figure 4, [41]). Alterations in the glutamatergic input of climbing fibers onto PCs have shown to be altered in SCA1 mice [85,86]. Deficits in CF-PC synaptic transmission have been observed in B05 transgenic mice at 6 weeks of age [87]. Furthermore, mutant ataxin-1 markedly increases inhibitory inputs from basket cells onto the PCs during cerebellar development, thereby disrupting cerebellar PC function [88]. In addition to alterations in synaptic input, the intrinsic activity of PCs can determine the PC output, which has been shown to be altered in several SCAs [41]. PCs are able to produce spontaneous electrical activity in the absence of synaptic input. This autonomous spiking is dependent on the proper function of a number of potassium channels that are highly expressed in the dendritic membrane of PCs [83,84,89]. For instance, voltage-gated potassium channels can modulate intrinsic firing, and the calcium-dependent potassium channels are involved in maintaining the autonomous spiking of PCs [41]. Alterations in PC intrinsic firing have been observed in several SCAs, including SCA1, where a loss of spontaneous cerebellar PC spiking is observed [41]. This is caused by a disruption of calcium homeostasis, leading to reduced function of the large conductance calcium-activated potassium (BK) channels. Additionally, a dysfunction in other potassium channels, such as voltage-gated potassium channels, can lead to higher A-type potassium currents (Ika) [82,83,84,89,90]. This dysfunction may contribute to dendritic hyperexcitability, thereby eventually contributing to neurodegeneration [84,90]. 

Next to the potassium channel dysfunction, contributing to alterations in the PC intrinsic activity, altered excitatory glutamatergic inputs, linked to the type-1 metabotropic glutamate receptors (mGluR1), also contribute to the altered PC activity observed in SCA1 (Figure 3, [91,92,93]). Reductions in mGluR1 mRNA levels as well protein levels have been observed in symptomatic elderly SCA1^154Q/2Q^ mice [91,94]. Furthermore, in pre-symptomatic B05 transgenic mice, reduced mGluR-excitatory postsynaptic current (EPSC) amplitudes were observed, leading to altered PC activity and eventually to PC death [93]. These results suggest that a loss of mGluR1 in PCs is associated with the pathological phenotype of SCA1 [91,93]. Contradictorily, another study showed prolonged mGluR1 currents, despite reduced EPSC amplitudes, in cerebellar parallel fiber synapses in symptomatic B05 mice, which are thought to be caused by a loss of glutamate transporter activity [95]. However, the majority of studies indicate a reduction in mGluR1 activity. Additionally, it has been shown that genes encoding for mGluR1 signaling proteins, such as Homer-3, are downregulated in B05 mice cerebellar PCs, which can lead to neuronal dysfunction in SCA1 [86,91,96]. As PC firing dysfunction is a common pathologic phenomenon in multiple SCAs, therapies restoring this dysfunction by targeting changes in the synaptic input as well as the intrinsic firing of PCs might be applicable to treat multiple ataxias [41]. The SCA-induced alterations in both potassium and glutamate receptors could be targeted by several drugs [97].

#### 3.2.1. Potassium Channels

To restore PC spiking in several SCAs, BK channel function could be restored, and IKa currents could be reduced. BK channel function can be restored either by enhancing BK channel expression or by activating the BK channels that are present (Figure 3, [83,84,90]). Dual administration of both stereotactically injected BK-AAV and intraperitoneally injected baclofen, a GABAb receptor agonist that potentiates a sub-threshold-activated potassium channel current in PCs, was performed. Treatment effectively normalized the dendritic excitability in the PCs, rescued neuropathology, and, moreover, significantly improved motor coordination two weeks after initiation of therapy in B05 transgenic mice (Table 3, [83,84,90]). BK channels can also be activated through pharmacological targeting with chlorzoxazone, which is a known activator of calcium-activated potassium channels. Dual treatment with chlorzoxazone and baclofen, dissolved in drinking water, normalized dendritic excitability in the PCs and significantly improved motor performance of B05 as well as in SCA1^154Q/2Q^ mice in a stage where motor impairment is cerebellar in origin [84,90]. On the other hand, at 20 weeks of age, when motor dysfunction is caused by motor neuron rather than cerebellar dysfunction, dual treatment with chlorzoxazone and baclofen did not rescue motor dysfunction in the SCA1^154Q/2Q^ mice [90]. As chlorzoxazone and baclofen are both muscle relaxants, they have tolerability concerns in patients with neurological disorders and older adults [90,98]. However, retrospective SCA patient data show that co-administration is tolerated and may improve SCA symptoms, suggesting that this treatment might be promising for use in clinical trials [90].

Another therapeutic strategy that targets voltage-gated potassium channels reduces the observed enhanced IKa currents in SCA1 pathology. Aminopyridines (APs) are potassium channel blockers with a high affinity for A-type potassium channels. Subcutaneous injection of APs has been shown to normalize the firing frequency of PCs and the motor dysfunction of early symptomatic B05 mice [82]. However, no effects were observed when treating B05 mice with an advanced SCA1 phenotype, suggesting that motor dysfunction at later stages is caused primarily by PC atrophy and death and not by electrophysiological dysfunction [82]. Surprisingly, starting subcutaneous injections with 3,4-diaminopyridine in an early stage of SCA1 delay the onset of motor dysfunction and partially prevent neurodegeneration in B05 mice [82]. This emphasizes the importance of starting treatment early in the disease process or even before the onset of disease symptoms. 

#### 3.2.2. Glutamatergic Signaling

The observed reduction in mGluR1-mediated EPSC amplitude can be modulated by so-called positive allosteric modulators (PAMs), which amplify the mGluR1 receptor function [91,94,95]. Subcutaneous injection of SCA1^154Q/2Q^ mice with the Ro0711401 PAM led to a significant increase in motor performance after 30 min [91]. Surprisingly, this effect on motor performance lasted for 6 days, which was well beyond the time of drug clearance from the cerebellar tissue. This could be explained by the induction of cerebellar long-term depression during the repeated execution of the rotarod assay, which could possibly induce a form of procedural memory [91]. Additionally, defects in learning and memory were improved [94]. On the other hand, intraperitoneal injection of the JNJ16259685 negative allosteric modulator (NAM) in SCA1^154Q/2Q^ mice resulted in markedly reduced motor function [91]. Contradictorily, one study found an improvement in motor performance after subcutaneous injections in conditional B05 transgenic mice with the same NAM [95]. 

Furthermore, the modulatory effects of baclofen on mGluR1 have been tested [93]. Subarachnoid injections of baclofen enhanced cerebellar mGluR1 signaling and improved the motor performance in B05 mice for approximately 1 week [93]. It is suggested that baclofen can improve motor functions in the transgenic mice either through interacting with the mGluR1 and enhancing mGluR1 signaling through the secretion of brain-derived neurotrophic factor (BDNF) or through activation of insulin growth factor 1 (IGF-1) receptors [93]. However, as previously mentioned, baclofen could additionally exert its effects through potentiation of sub-threshold-activated potassium channel currents in PCs [90].

In addition to targeting mGluR1 directly, mGluR1-related signaling molecules can be targeted to ameliorate SCA1 pathology. For instance, a reduction in Homer-3, which is an adaptor protein of mGluR1, has been observed in early symptomatic SCA1^154Q/2Q^ mice [85,93]. This reduction is likely mediated through reduced RORα-mediated transcriptional activity caused by mutant *ATXN1* [93]. Homer-3 binds to mGluRs, and its reduction leads to altered PC activation responses [85,93]. To improve these altered PC activation responses, Ruegsegger and colleagues enhanced Homer-3 expression using an AAV [85]. AAV-Homer-3 treatment in SCA1^154Q/2Q^ mice ameliorated PC climbing fiber deficits, rescued dendritic spine loss, and improved behavioral outcomes [85]. 

#### 3.2.3. NMDA Receptor

Although there is no convincing evidence that the extra-synaptic *N*-methyl-D-aspartate receptor (NMDAR) is involved in SCA1, administration of memantine, which is a NMDAR antagonist that preferentially inhibits the extra-synaptic NMDAR, had beneficial effects (Figure 3, Table 3, [99]). Oral treatment with memantine attenuated body weight loss and extended the life span of treated SCA1^154Q/2Q^, possibly though preventing neuronal cell death in the brainstem or PC death in the cerebellum [99]. This suggests that extra-synaptic NMDAR contributes to the pathogenesis of SCA1 and that treatment with memantine could ameliorate this [99]. 

#### 3.2.4. Protection of PC Survival and Function

Next to restoration of the PC electrical activity, several other pharmacological therapeutics have been developed that aim to protect PC survival and PC function. For SCA1, IGF-1, vascular endothelial growth factor (VEGF), BDNF, and lithium carbonate have been tested in mouse models (Figure 5, Table 3). IGF-1 is a trophic factor for both glia and neurons in the cerebellum, and it promotes PC survival and dendritic growth. IGF-1 and its receptors are also present in the climbing afferents of the inferior olive nucleus, where it is involved in motor learning processes, thereby influencing PC synaptic activity [100]. It has been found that impairments in the IGF-1 pathway are involved in neurodegenerative processes [112]. Intranasal administration of IGF-1 in B05 mice resulted in a dose-dependent improvement in motor performance as well as the restoration of the calcium buffer protein calbindin-D28K and protein kinase C-γ (PKC-γ) expression, which are both reduced in SCA1 [100]. It is suggested that restoration of these proteins could lead to improved calcium homeostasis and PKC-γ-mediated signaling in SCA1 PCs [100].

VEGF is another trophic factor, which additionally functions as an angiogenic factor. VEGF is widely expressed in PCs in the cerebellum and is downregulated in SCA1^154Q/2Q^ mice as well as human patients, possibly through a repressive effect of the mutant ataxin-1 [101,102]. Reduced levels of cerebellar VEGF in SCA1 lead to a decrease in microvessel density and hypoxia as well as a decrease in the growth and survival of PCs [101]. To increase VEGF expression, pharmacological delivery of recombinant VEGF (rVEGF) and a synthetic VEGF mimetic peptide (nano-VEGF) were tested in SCA1^154Q/2Q^ mice models [101,102]. ICV administration of rVEGF and nano-VEGF improved motor performance and resulted in a reduction in neuropathology in SCA1^154Q/2Q^ mice [101,102]. However, nano-VEGF administration outperformed rVEGF therapy in improving the levels of capillary proteins and amelioration of microvascular health [102]. These improved outcomes of nano-VEGF delivery could possibly be assigned to its increased biological stability compared with rVEGF amongst others [102].

Brain-derived neurotrophic factor (BDNF) is an important trophic factor for cognitive and motor function and plays key roles in both survival signaling and neuroplasticity. BDNF levels are reduced in the cerebellum and medulla of SCA1 patients [103]. Furthermore, BDNF levels have shown to be decreased in the cerebellum of transgenic B05 mice, and treatment with BDNF improved cerebellar pathology and motor impairments in these mice [103,113,114]. Subcutaneous BDNF treatment in SCA1^154Q/2Q^ mice additionally showed amelioration of motor and cognitive deficits as well as improved cerebellar and hippocampal pathology [103].

Lithium carbonate has an effect on several cellular functions, and it has shown to exert neuroprotective effects, possibly by affecting gene transcription or by reducing oxidative stress (Figure 5, [104,105,115,116]). Furthermore, lithium carbonate has been shown to suppress neurodegeneration by partially rescuing cell death in SCA1^154Q/2Q^ mice [104]. Administration of 0.2% lithium carbonate in the chow of SCA1^154Q/2Q^ mice resulted in a rescue of motor function as well as spatial learning and memory impairment [104]. Treatment also partially rescued dendritic pathology and reduced the GSK3-ß activity, thereby possibly inhibiting apoptosis [104]. Moreover, lithium carbonate was shown to regulate purine, oxidative stress, and energy production metabolic pathways [105]. Although is not clear how lithium carbonate exerts neuroprotective effects, this could be mediated through apoptosis inhibition, restored purine metabolite levels, improvement in transcriptional dysregulation, or by affecting neurotransmission or neurogenesis [104,105].

#### 3.2.5. Mitochondrial Functioning

PCs have a high metabolic demand due to their synaptic trafficking of several proteins and vesicles as well as for the maintenance of the resting membrane potential [106,107]. Because neurons have limited glycolytic capacity, they are highly dependent on oxidative phosphorylation (OXPHOS) in the mitochondria, which is a major producer of reactive oxygen species (ROS) [106]. OXPHOS dysfunction and oxidative damage has been observed in the PCs in SCA1^154Q/2Q^ mice [106,107]. Therefore, targeting the OXPHOS dysfunction or ROS could ameliorate SCA1 disease pathology (Figure 5, Table 3). 

Treatment with MitoQ, a mitochondria-targeted anti-oxidant, restored mitochondrial functioning and ameliorated disease symptoms by reducing oxidative stress in SCA1^154Q/2Q^ mice [106]. Furthermore, in B05 transgenic mice, OXPHOS complex I deficits were observed [107]. Targeting this dysfunction with succinic acid boosts cerebellar function in these animals. Succinic acid is thought to bypass the dysfunctional complex I without disrupting complex II or downstream oxygen consumption in the OXPHOS [107]. It additionally restores complex III inhibition and prevents damage of the outer mitochondrial membranes. In addition to amelioration of mitochondrial function, succinic acid treatment reduced PC dendritic atrophy and the loss of PCs accompanied by a less severe cerebellar ataxia phenotype [107]. 

#### 3.2.6. DNA Damage Repair, Transcription, and Replication

Through the altered interaction of mutant *ATXN1* with several transcription and splicing factors, transcriptional dysregulation occurs [39,40]. Furthermore, it is thought that DNA damage repair might also be impaired in SCA1 pathology [108,109]. Several studies have tried to target the transcriptional dysregulation as well as the DNA damage repair pathway, which are involved in SCA1 pathology (Figure 5). 

In SCA1, the expression of mutant *ATXN1* was shown to reduce high-mobility group box 1 (HMGB1), thereby inhibiting damage repair [108]. HMGB1 is a chromatin protein which regulates the higher structure of genomic DNA, thereby facilitating the binding of other proteins and influencing transcription and DNA damage repair [106,108]. To rescue the imbalance of gene expression observed in SCA1, HMGB1 was used as a therapeutic strategy [108]. Restoring HMGB1 expression in SCA1^154Q/2Q^ mice with an AAV-based method (AAV1-HMGB1) showed an increase in lifespan and improvement in motor activity [108]. However, the SCA1 phenotype could not be completely normalized after HMGB1 treatment, suggesting that there was insufficient upregulation of HMGB1 or that other mechanisms also contribute to SCA1 pathology [108]. Furthermore, there are several concerns for the therapeutic usage of HMGB1, including inflammation induction and the broad effect of HMGB1 on general gene expression, which might induce unwanted side effects in cells [108].

RpA1 is another molecule that is involved in multiple DNA damage repair pathways, including homologous recombination and non-homologous end joining (Figure 5). Genetic screens in *Drosophila* as well as in silico models have shown that RpA1 might be involved in DNA damage repair in SCA1 pathology [109,117]. Therefore, a gene therapy with AAV-RpA1 was tested in SCA1^154Q/2Q^ mice. Treatment with AAV-RpA1 showed a recovery of motor function, DNA damage, dendritic shrinkage, and spine abnormalities [109]. It is suggested that these abnormalities are resolved through decreasing the DNA damage in the affected neurons and PCs [109]. However, RpA1 might also ameliorate disease symptoms by affecting RNA splicing, transcription, and the cell cycle [109].

#### 3.2.7. The Proteostatic Machinery

As SCA1 is a protein aggregation disease, therapeutics that target the proteostatic machinery to promote protein breakdown and prevent aggregation have also been tested. P21-activated kinases (PAKs) are a family of serine/threonine kinases (Figure 5, Table 3, [110]). A genetic screen in a fly SCA1 model has shown that PAK1 might be a potential modulator of *ATXN1* [110]. It was suggested that PAK1 can regulate *ATXN1* clearance through interfering with the proteasome-mediated degradation, possibly by inhibiting a ubiquitin E3 ligase or other factors in the proteasome pathway that normally promote ubiquitination and degradation of *ATXN1*. Indeed, pharmacological inhibition of all PAKs resulted in a reduction in *Atxn1* levels in SCA1^154Q/2Q^ mice, suggesting that PAK1 regulates *ATXN1* stability and toxicity [110]. 

#### 3.2.8. Inflammation

As gliosis correlates with disease severity in SCA1 patients, and microglia are activated in several mouse models of SCA1, therapeutic strategies targeting microglia are also studied for SCA1 [23,118]. Colony-stimulating factor 1 receptor (CSF1R) signaling is essential for microglial survival [111]. Therefore, pharmacological inhibition of the CSF1R signaling through administration of PLX3397 can be used to cause microglial depletion after several days of administration (Figure 5, Table 3, [111]). Administering PLX3397 in in pre-symptomatic B05 mice resulted in a 69% decrease in microglial density and improved motor function, although no significant alterations in PC atrophy, synaptic loss, or altered astrogliosis were observed [111]. It is suggested that the improved motor function after treatment could be caused by an increased expression of post-synaptic density 95 (PSD95) and wildtype *ATXN1* as well as the reduction of the pro-inflammatory cytokine TNF-α [111]. Increased PSD95 expression may preserve the functions of the remaining synapses, whereas increased levels of wild-type *ATXN1* promote the function of PCs, thereby possibly being protective in SCA1.

### 3.3. Stem Cell Replacement Therapies

Instead of targeting either *ATXN1* or other molecules involved in disease pathology, another therapeutic strategy aims to replace the cells that undergo neurodegeneration in SCA1 using stem cell replacement [119]. Several stem cell types can be used for replacement therapies, including pluripotent stem cells (PSCs) and multipotent stem cells, such as neural progenitor cells (NPCs) and mesenchymal stem cells (MSCs) (Table 4, [21,119,120,121,122,123,124,125]). 

MSCs are multipotent and reside in several tissues but mainly in the bone marrow and fat, where they support hematopoiesis and produce cells of the mesodermal lineage. Additionally, they also have immunomodulatory and neurotrophic properties [119]. The underlying mechanism of the therapeutic effects of MSCs is not yet fully understood, but it is thought to occur through cellular replacement, fusion with degenerating neurons, or neuronal trophic support [119,120]. Pre-clinical studies using MSCs have been tested in both the B05 SCA1 transgenic as well as the SCA1^154Q/2Q^ mouse model [119]. Results from different pre-clinical studies using B05 transgenic mice show similar results [120,121,122]. IT administration of MSCs in pre-symptomatic B05 mice results in a thicker granular layer, rescue of dendritic pathology, and improvements in motor coordination [121]. Another study found that transplantation of MSCs isolated from the Wharton’s jelly of the umbilical cord directly into the left and right cerebellar cortex of pre-symptomatic transgenic B05 mice improved cerebellar atrophy as well as motor and behavioral deficits [122]. These ameliorating effects of MSC therapy were not caused by a direct differentiation of the MSCs into neurons or astrocytes after transplantation [122]. This suggests that the neuroprotective effects observed are possibly attributed through neurotrophic factors [121,122]. On the other hand, a study injecting human fetal MSCs into the cerebellum of symptomatic transgenic B05 mice showed fusion of the MSCs with PCs and molecular layer interneurons, thereby rescuing cerebellar degeneration [120]. However, this fusion was not observed in non-symptomatic SCA1 mice or wildtype mice, suggesting that fusion occurs only when degeneration is ongoing [120]. 

In addition to transgenic B05 mice, MSC therapies were also tested in SCA1^154Q/2Q^ mice to assess the effects of these therapies on the PNS, as motor neuron degeneration in this area is observed in SCA1 patients. IT administration of GFP-labeled MSCs into the subarachnoid space in pre-symptomatic SCA1^154Q/2Q^ mice showed distribution of the MSCs into the ventral root of the spinal cord, which resulted in larger axons and bigger myelin sheaths compared with non-treated SCA1^154Q/2Q^ mice [21]. In this study, administration of MSCs was already efficient in a dose 10 to 100 times lower than the dose used in human studies, suggesting that lower doses might be sufficient to reach therapeutic effects in SCA1 patients, although this will need to be confirmed in human clinical trials [21]. These results might also suggest that the neuronal degeneration is more likely rescued due to trophic factors released from MSCs and not due to the direct differentiation of MSCs into neurons [21]. In another study, administration of MSC-conditioned medium, was tested in SCA1^154Q/2Q^ mice [123]. Mice either received a single IT injection at 4 weeks of age or they received intravenous injections every 2 weeks from 4-12 weeks of age. Both IT and intravenous injections from the pre-symptomatic stage onwards improved the progressive degeneration of both axon and myelin in the spinal neurons, supporting the notion that therapy is effective mainly through the secretion of neurotrophic factors [123]. Both administration routes resulted in improved motor coordination, although repetitive intravenous injections resulted in reduced rotarod performance, suggesting that a single IT injection might be safer than multiple intravenous injections [123].

Next to MSCs, NPCs can also be used as a cell source for stem cell therapy [124,125]. NPCs derive from the ectoderm and can give rise to neuronal and glial cell populations of the CNS. NPCs either can be directly isolated from primary CNS tissue or can be differentiated in vitro from ESCs or iPSCs using specific growth factors [119]. The therapeutic effect of NPCs could be due to cell replacement or differentiation into oligodendrocyte progenitor cells, thereby allowing remyelination or through trophic support [119]. As the transplantation of NPCs has proven to be safe and effective in Parkinson’s disease patients, its translation into the clinic seems feasible [119,126]. In a pre-clinical study, embryonic 13–15 day primordium NPCs were isolated from mice and subsequently injected into the deep cerebellar nuclei of 6- to 9-week-old, early symptomatic, transgenic B05 SCA1 mice [125]. Transplantation led to improved motor performance and amelioration of PC dendritic morphology [125]. Despite the ongoing neurodegeneration in the SCA1 mice brains, the donor cells survived and elicited a beneficial effect in the mice up to 20 weeks [125]. Another pre-clinical study transplanted mouse NPC derived from the adult subventricular zone into the cerebellar white matter of transgenic symptomatic B05 SCA1 mice [124]. Transplantation in the symptomatic SCA1 mice improved motor coordination and improved PC survival, function, and morphology [124]. However, treatment in pre-symptomatic SCA1 mice, when PC degeneration was minimal, did not show any improvements, suggesting that the time of treatment is crucial [124].

Pre-clinical studies have shown that replacement therapies using MSCs and NPCs can improve neuropathology and motor coordination in SCA1 mice [121,122,124,125]. Because MSCs can be more rapidly expanded ex vivo compared with NPCs, it is easier to obtain sufficient MSC numbers to treat a large number of patients [119,120,122]. Additionally, the self-renewal capacity and differentiation of MSCs give them great therapeutic potential, although caution should be taken, as passaging the MSCs for 20 times or more might acquire a similar profile to Ewing’s sarcoma [120,127]. Furthermore, through the immunomodulatory actions of MSCs, they have low immunogenicity and can avoid host immune responses, thereby allowing for allogenic engraftments [119,127]. Therefore, several phase I/II clinical trials using MSCs have been performed on SCA1 patients, which are discussed below [127,128]. 

## 4. Clinical Trials

As outlined in this review, several therapeutics for SCA1 have been tested in pre-clinical animal models. Therapies focus either on modulating the levels of ataxin-1, restoring the normal function of several targets involved in SCA1 pathophysiology, or replacing the cells that have been lost due to neurodegeneration. Pre-clinical studies have shown promising results for these strategies, and some therapies were additionally tested in clinical trials [127,128]. 

Most clinical trials in SCA1 have focused on restoring the PC activity by targeting the potassium channels and the glutamatergic channels. A clinical, open label trial with the slow-release form of 4-AP (Dalfampiridine) was performed in 16 patients with chronic cerebellar ataxia, including 3 patients with SCA1 [129,130]. Treatment induced modest, short-term improvements in ataxia [129]. A long-term clinical, placebo-controlled cross-over trial (NCT01811706) with Dalfampiridine was additionally performed in 20 patients with SCA types 1, 2, 3, or 6 [131]. However, results showed no significant improvement of ataxia or other measures of gait [131].

Other clinical studies have been performed to improve the electrical function of PCs by using Riluzole, Troriluzole, or CAD-1883, which all act by promoting the opening of small-conductance calcium-activated potassium channels and by enhancing glutamate transporters [132]. A randomized, double-blind, placebo-controlled pilot trial with Riluzole was performed in 40 patients with cerebellar ataxias, including two SCA1 patients [133]. Results showed that treatment with Riluzole improved the ICARS score in patients [133]. These results were confirmed in a 1-year randomized, double-blind, placebo-controlled trial with Riluzole in patients with several types of genetic ataxia, including SCAs, showing a higher proportion of improved SARA scores in the Riluzole-treated group [134]. However, a recent similar trial in SCA2 patients showed neutral results (NCT03347344) [135]. Troriluzole, a pro-drug of Riluzole, was studied in a clinical trial in a cohort of SCA patients, including 35 SCA1 patients (NCT02960893). Subsequently, a phase 3 clinical trial using this drug in SCA has been performed, although results have not yet been published (NCT03701399). On the basis of these trial results, it is not clear whether Riluzole, and variants thereof, could possibly be a treatment option for patients with SCA [136]. Furthermore, CAD-1883, an allosteric modulator of the small-conductance calcium-activated potassium channels, has been investigated [132]. An open-label phase 2 clinical trial with CAD-1883 has begun for patients with SCA, although results still need to be published (NCT03688685). 

Additionally, oral lithium has been investigated in an open-label, non-randomized phase I trial (NCT00683943), but results have never been published [7]. Moreover, in collaboration with VICO therapeutics, the (CUG)^7^ ASO (VO659) has recently entered a phase I/II open-label clinical trial, including 95 SCA1, SCA3, and Huntington’s disease patients to investigate the safety, tolerability, pharmacokinetics, and pharmacodynamics of ascending doses of intrathecally administered V0659 (Clinical Trials Register).

In one open label clinical study, 14 SCA patients were IT injected with umbilical cord MSCs for 4 weeks [127]. Treatment resulted in significantly improved ataxia severity and quality of life, as assessed by the ICARS and ADL scores, respectively, without any serious adverse effects [127]. In another open-label, uncontrolled phase I/II clinical study, a combination of umbilical cord MSC intravenous injections and IT injections were tested [128]. Treatment resulted in significantly better improvements in the Berg Balance Scale compared with baseline [128]. Additionally, ICARS scores improved in the 3rd and 6th month after treatment, suggesting that the MSC therapy had an effect on ataxia severity. 

Although results from these clinical studies on MSC transplants seem efficient and safe, a meta-analysis of these studies showed no significant differences in the ICARS scores before and after treatment, low certainty in estimates of quality, and a high risk of bias, as both studies were uncontrolled clinical trials [137]. Furthermore, in general, a lack of standardization in the used protocols, dosage, and type of administration in pre-clinical studies might slow down the translation of SCA1 therapies into the clinic [119]. To be able to assess the efficiency of the different therapies in SCA patients, more clinical studies with a larger sample size and robust design should be performed [137].

## 5. Future Perspectives on SCA1 Therapeutics

Although several human clinical trials have been performed, no therapies for SCA1 have reached market authorization yet. For future SCA1 therapeutics to reach the clinic, several challenges, mainly in the translation of preclinical findings into clinical trials and in clinical trial design, need to be overcome [138,139]. Mouse models are essential to study biodistribution and bioavailability as well as the effect of a potential treatment on the ataxia phenotype. However, they either overexpress the mutant ataxin-1 protein, or the mutant protein is expressed only in a subset of brain cells. Furthermore, there are considerable differences between the mouse and human genomes, making translation of findings from animal models to patients often difficult [27,28,48]. Including models from patient-derived induced pluripotent stem cells (iPSCs) into the pre-clinical drug-development pipeline might improve translation into clinical research, as these reflect endogenous protein expression levels as well as the patient’s genetic background [138,140]. 

Because of the BBB, delivery of drugs to the brain is a major challenge. Finding the right administration route and dose of a drug in order to reach a therapeutic dose in the brain without inducing toxicity, remains challenging and should be assessed for each therapeutic strategy [54]. Another issue is determining the stage of the disease process at which treatment should commence. As SCA1 is a monogenetic disorder, the cause of the disease is present from birth, whereas symptom onset usually occurs later in life, when neurodegeneration already occurred. It will be critical to determine at what stage of the disease progression pathology is irreversible and at what stage it can still be slowed down or maybe even halted [141,142,143]. Currently, the presence of symptoms is an inclusion criteria for clinical trials, and maybe neurodegeneration at this stage of the disease is already too far advanced, reducing the chances of showing a treatment effect within the relatively short duration of a trial. However, the inclusion of non-symptomatic patients in clinical trials might raise ethical concerns. Future research into biomarkers that can accurately detect changes in SCA1 patients well before onset of clinical symptoms will be crucial to starting clinical trials at earlier disease stages when neurodegeneration is less advanced and the likelihood of beneficial effects of the treatment are higher [141,144]. 

Drug development is a long and expensive process, and as SCA1 is a relatively rare disease, it is less attractive for industry to invest in SCA1 drug development [54]. Therefore, it is important to evaluate the costs and benefits of SCA1 therapeutics [145]. Therapy costs can be reduced by slowing progression of SCA1 in early disease stages and avoiding treating disease in later, more severe stages [145]. Moreover, it is difficult to recruit sufficient patients into clinical trials and to show significant treatment effects with small group sizes [144,146]. To overcome this problem, improvements in clinical trial designs could allow for the inclusion of smaller numbers of patients. For instance, multiple drugs could be tested against a single placebo arm, or a cross-over design could be used, thereby reducing the number of patients that need to be included in the clinical trial [144]. Moreover, drugs that target common pathogenic mechanisms observed in multiple SCAs, such as the altered PC activity, could be beneficial [41]. There is a world-wide increase in awareness that new and innovative solutions are needed for the development of treatments for rare diseases [147].

In SCA1, disease progression is slow, and clinical rating scales might not be sensitive to measure effects in trials that run for one year [4,146]. Improving and unifying these clinical rating scales will reduce variability in clinical trial outcome measures. The variability in disease severity, age of onset, and symptom progression among SCA1 patients further complicates the accurate measurement of the response to potential therapies [4]. An integrated staging system, such as recently proposed for Huntington’s disease, would reduce variability in clinical data acquisition but could also facilitate the inclusion of patients before the onset of clinical signs and symptoms of ataxia [148]. Additionally, natural history data are crucial to better understand SCA1 disease progression and to identify biomarkers and factors that influence the disease course [144,149,150]. Several natural history consortia have been set up, including the READISCA and the Dutch SCA1 study [151]. 

SCA1 is a devastating disease, and to date only symptomatic treatment is available. With the advancements made in understanding SCA1 pathology and the progression of different intervention strategies, there is great promise that a therapy to treat SCA1 patients will become available in the coming years. There are some pharmacological treatments targeting dysregulated pathways in SCA1 as well as promising strategies modulating ataxin-1 levels, with several RNA targeting therapies for SCA1 in advanced stages of preclinical development. Another promising strategy for SCA1 is gene editing using CRISPR/Cas9, which has already successfully been tested in a pre-clinical study in hiPSCs [62,152]. However, before CRISPR/Cas9-based gene editing can be evaluated in human clinical trials, limitations regarding off-target effects and the use of viral vectors for delivery to the brain will first have to be resolved [152,153]. Additionally, future treatment strategies might involve a combination of treatments both targeting the polyglutamine ataxin-1 as well as other dysregulated pathways that are further downstream in the SCA1 disease pathology. Additionally, combinatorial treatments might be necessary to target region-specific regulators of ataxin-1. For a rare disease such as SCA1, worldwide collaborations are essential to move forward to gather, standardize, and share data towards successful clinical development of a therapy for SCA1.

## Figures and Tables

**Figure 1 biomolecules-13-00788-f001:**
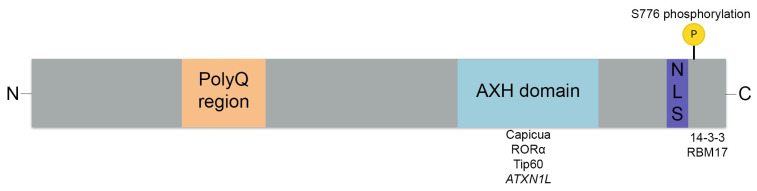
Ataxin-1 protein and its functional domains. The ataxin-1 protein with the polyglutamine (polyQ) region, the AXH domain, and the C-terminal domain containing the NLS and the S776 phosphorylation site. Binding partners of these domains are depicted under their respective domains.

**Figure 2 biomolecules-13-00788-f002:**
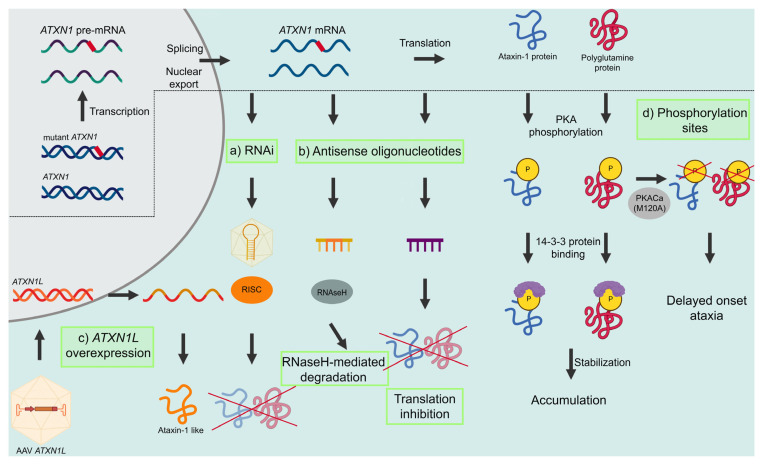
Ataxin-1 modulating therapies. Ataxin-1 modulating therapies that are currently being investigated as possible therapeutics for SCA1. These strategies include (**a**) RNAi strategies, which involve binding to the RISC, (**b**) ASOs, including ASOs that mediate RNaseH-mediated degradation and translation inhibition, (**c**) ATXN1L, which can be overexpressed to preserve wildtype protein function, or (**d**) the S776 phosphorylation site, which can be targeted to reduce ataxin-1 levels.

**Figure 3 biomolecules-13-00788-f003:**
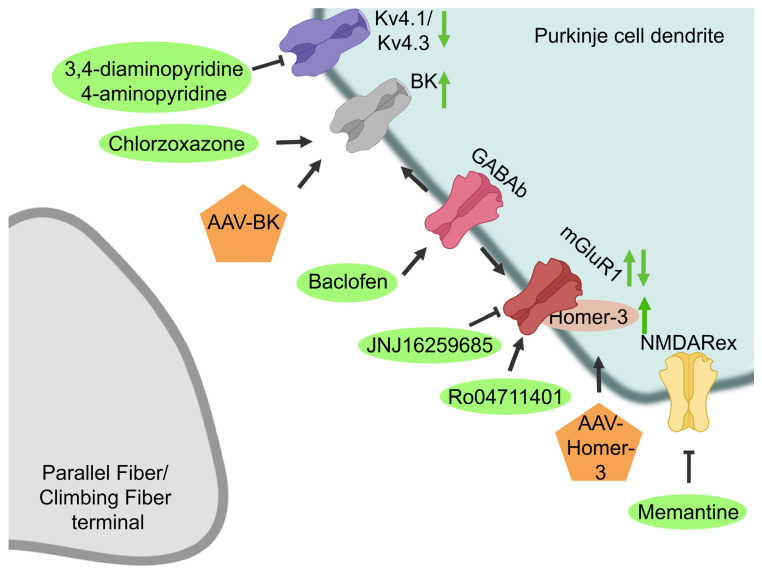
Receptor-mediated therapies. Cerebellar PC activity is mediated through several receptors and ion channels. In SCA1, cerebellar PC activity is dysregulated and can be restored through several drugs targeting either the potassium channels (Kv4.1/Kv4.3 and BK channels) or the mGluR1, NMDARex, or GABAb receptors. Green arrows indicate the effect of the therapies on the receptor and ion channel function.

**Figure 4 biomolecules-13-00788-f004:**
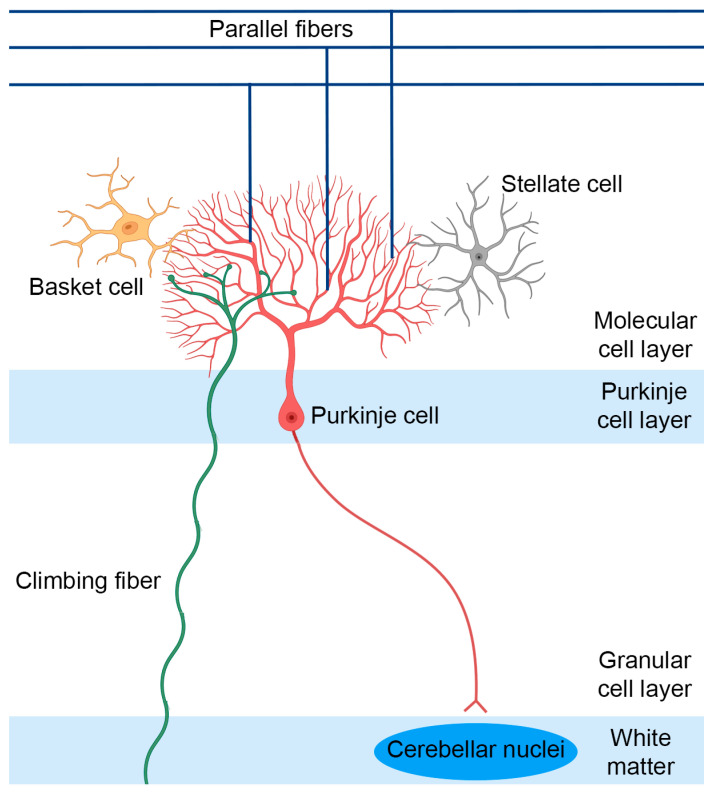
Overview of the cerebellar structure. The cerebellum consists of three layers, namely the granular cell layer, the purkinje cell layer, and the molecular cell layer. The PCs receive excitatory synaptic input from the CF and PF as well as inhibitory input from the basket and stellate cells.

**Figure 5 biomolecules-13-00788-f005:**
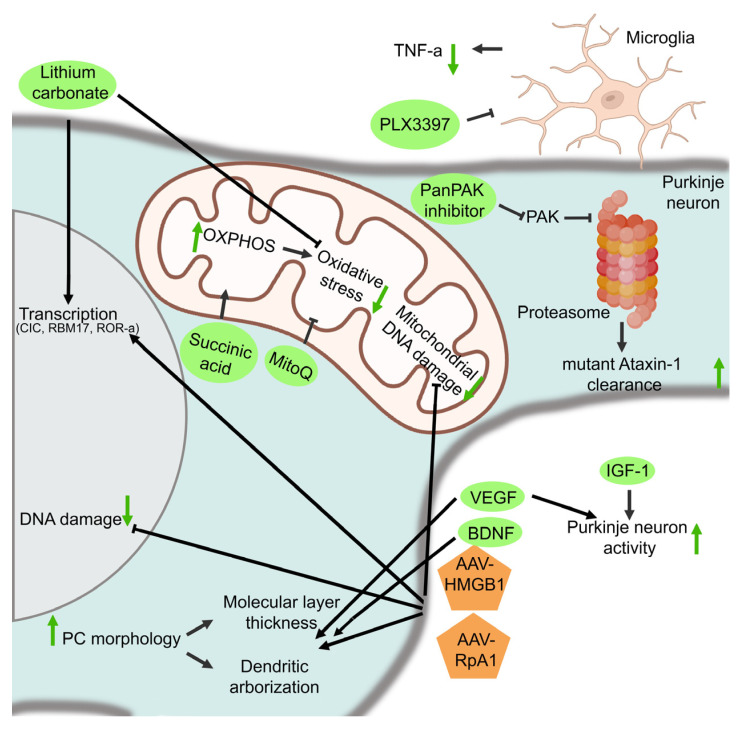
Other pharmacological targets for SCA1: These aim to restore PC survival and function, improve mitochondrial function, increase DNA damage repair, and enhance mutant ataxin-1 clearance by the proteasome or lower inflammation. Green arrows indicate the effect of the therapies on the different cell functions that are targeted.

**Table 1 biomolecules-13-00788-t001:** Animal models used in the pre-clinical development of SCA1 therapeutics.

SCA1 Animal Model	Genetic Background	Advantages	Disadvantages
SCA1 transgenic B05 mice	Human *ATXN1* gene containing 82 CAG repeats under control of a pcp2 promoter	Mimics cerebellar component of SCA1, including ataxia and classical neuropathological changes. Contains human *ATXN1*.	Expression only in the PCs of the cerebellum. Has 50 to 100 times overexpression of *ATXN1* compared with endogenous levels [27,48].
SCA1 knock-in mice (SCA1^154Q/2Q^)	154 CAG repeats in the mouse *Sca1* locus under control of endogenous promoters	Expresses endogenous *Atxn1* levels. Shows development of slow progressive neurodegeneration, ataxia, and deficits in memory (reflecting extra-cerebellar pathology).	Nuclear inclusions were found in more brain regions in the mice than were found in humans. Model contains very long CAG repeats, not in the range of SCA1 patients [28].
SCA1 *Drosophila melanogaster* (fly) model	Expresses human *ATXN1* gene containing 82Q in the eye retina using the GAL4/UAS system	Shows nuclear inclusion formation in the eye photoreceptor cells and CNS neurons as well as degeneration of the retina and neurodegeneration.	The model lacks a cerebellum [29].

**Table 2 biomolecules-13-00788-t002:** Summary of the different therapeutic strategies aimed at modulating ataxin-1 levels.

Mechanism	Model System	Molecular Findings		Pathology	Behavioral/Functional Tests
		RNA Level	Protein Level		
shRNA targeting human *ATXN1*	SCA1 transgenic B05 mice	-	-	Amelioration of PC pathology and rescue of nuclear inclusions in PCs.	Improvement in motor performance as assessed by the rotarod assay [55].
AAV expressing an artificial miRNA targeting human ataxin-1 (miS1)	SCA1 knock-in mice (SCA1^154Q/2Q^); pre-symptomatic	Up to 30% reduction	58 to 72% reduction of ataxin-1 levels	Amelioration of PC pathology and no glial activation.	Improvement in motor performance as assessed by the rotarod assay and improvement in gait analysis [58].
	SCA1 transgenic B05 mice; pre-symptomatic	Up to 70% reduction		Amelioration of PC pathology and no glial activation.	Improvement of motor performance as assessed by the rotarod assay [57].
	SCA1 transgenic B05 mice; symptomatic	Up to 67% reduction		Amelioration of PC pathology. Mice treated with the highest dose showed no *ATXN1*-positive PCs but showed enhanced immunoreactivity.	Rescue of motor performance to wildtype levels as assessed by the rotarod assay in a dose-dependent manner; rescue in NAA/inositol ratios [59].
	Rhesus monkey	Up to 30% reduction in the left cerebellar hemisphere		Eight weeks post-injection, microglial activation and astrogliosis were enhanced in the left cerebellar cortex and deep cerebellar nuclei [60].	
	Rhesus monkey	Significant reduction of *ATXN1* in medial and lateral cerebellar cortex		Toxicity in deep cerebellar nuclei was observed; necrosis, demyelination, perivascular/leptomeningeal lymphoid infiltrates, increase in lesion severity, PC loss, and enhanced immunoreactivity.	Monkeys developed ataxia, tremor, head-tilt, and dysmetria 3 months after treatment [61].
ASO353	SCA1 knock-in mice (SCA1^154Q/2Q^)	Significant reduction of total mouse *Atxn1* in brainstem and cerebellum		Rescue of analysis of neurochemicals and transcriptional disease signatures. No unwanted side effects involving BACE1, CIC activity or reduction in neuronal precursor cells.	Improved motor performance at 28 weeks, as assessed by the balance beam test. Pre-symptomatic treatment improved motor performance on both the rotarod and balance beam as well as prolonged survival [62,63].
(CUG)^7^ VO659	SCA1 knock-in mice (SCA1^154Q/2Q^)	An exon 8 skip product of the *Atxn1* gene was detected	A significant dose-dependent reduction in polyglutamine ataxin-1 in several brain regions. Reduction up to 45% in cerebellum and up to 56% in brainstem [64]		
Combination of miS1 and vectors expressing human *ATXN1L*	SCA1 transgenic B05 mice	Significant increase in human *ATXN1L* expression and a significant reduction in human *ATXN1* levels		Improvement in autonomous transcriptional changes and partial restoration of glial activation.	Significant improvement in latency to fall on the rotarod assay [65]

**Table 3 biomolecules-13-00788-t003:** Overview of the different pharmacological therapeutic strategies for SCA1.

Mechanism	Model System	Pathology	Behavioral/Functional Test
BK-AAV and baclofen (intracerebellar injection)	SCA1 B05 transgenic mice	Mitigation of dendritic degeneration	Improves motor performance [83]
Chlorzoxazone and baclofen (oral administration)	SCA1 B05 transgenic mice	Rescue of dendritic excitability in PCs	Improves motor performance [84]
	SCA1 knock-in mice (SCA1^154Q/2Q^)	Rescue of dendritic excitability in PCs	Improves motor performance without affecting muscle strength [90]
4-aminopyridine and 3,4-diaminopyridine (subcutaneous injection)	SCA1 B05 transgenic mice	Normalization of firing rate of PCs; partial protection against cell atrophy	Improves motor performance [82]
mGlu1 receptor PAM (Ro0711401; subcutaneous injection)	SCA1 knock-in mice (SCA1^154Q/2Q^)	Increase in the number of dendritic spines	Significantly improves motor performance; effect lasted for 6 days [91]
Baclofen (intrathecal injection)	SCA1 B05 transgenic mice	Enhancement of mGluR1 signaling	Improves motor performance [93]
AAV-Homer-3 (ICV)	SCA1 B05 transgenic mice	Restoration of mTORC1 signaling and neuronal activation; improvement in PC morphology	Significantly improves motor performance [85]
Memantine (oral administration)	SCA1 B05 transgenic mice	Rescue of PC density; significant reduction in PC nuclear inclusions	Attenuates body weight loss and extends the life span [99]
IGF-1 (intranasal administration)	SCA1 B05 transgenic mice	Restoration of PC pathology	Improves motor performance [100]
Recombinant VEGF (ICV)	SCA1 knock-in mice (SCA1^154Q/2Q^)	Restoration of PC dendrite pathology and firing rate; improved microvascular health.	Improves motor performance [101]
Nano-VEGF (ICV)	SCA1 knock-in mice (SCA1^154Q/2Q^)	Restoration of PC dendrite pathology and firing rate; improved microvascular health.	Improves motor performance [102]
BDNF	SCA1 knock-in mice (SCA1^154Q/2Q^)	Improvement in cerebellar and hippocampal pathology	Ameliorates motor and cognitive deficits [103]
Lithium carbonate (oral administration)	SCA1 knock-in mice (SCA1^154Q/2Q^)	Partial rescue of dendrite pathology, restoration of isoprenylcystein carboxyl methyltransferase levels; reduction of GSK3ß levels; restoration of purine, oxidative stress, and energy production metabolic pathways	Improves motor performance, spatial learning, and memory [104,105]
MitoQ (oral administration)	SCA1 knock-in mice (SCA1^154Q/2Q^)	Amelioration of mitochondrial morphology and function; improvement in PC numbers and function	Improves motor performance [106]
Succinic acid (oral administration)	SCA1 B05 transgenic mice	Rescue of complex I OXPHOS dysfunction; amelioration of molecular layer and PC layer degeneration	Improves motor performance [107]
AAV1-HMGB1 (intrathecal injection)	SCA1 knock-in mice (SCA1^154Q/2Q^)	Rescue of PC loss; improvement in PC morphology and molecular layer thickness as well as nuclear and mitochondrial DNA damage	Improves motor performance and increased survival ratio [108]
AAV-RpA1 (intrathecal injection)	SCA1 knock-in mice (SCA1^154Q/2Q^)	Rescue of γH2AX and 53BP1 levels; recovery of mitochondrial DNA damage, dendritic pathology; rescue of molecular layer thickness, impaired splicing, transcription, and abnormal cell cycle	Improves motor performance [109]
panPAK inhibitor (intraperitoneal injection)	SCA1 knock-in mice (SCA1^154Q/2Q^)	Significant reduction in both expanded and wildtype *ATXN1* RNA levels in the cerebellum [110]	
PLX3397 (oral administration)	SCA1 B05 transgenic mice	Reduction in microglial density in the cerebellum; reduction in TNF-α and increase in PSD-95 expression; increase in WT ataxin-1 protein levels	Improves motor performance [111]

**Table 4 biomolecules-13-00788-t004:** Overview of the different stem cell replacement therapies investigated for SCA1.

Cell Type	Delivery Method	Delivery Location	Model	Treatment Age	Pathology Outcome
MSCs	Single IT injection	Subarachnoid space	SCA1 B05 transgenic mice	5 weeks, pre-symptomatic stage	Amelioration of PC and PC dendritic spine pathology as well as improved motor coordination [121]
Human fetal MSCs	Single intracranial injection	Cerebellar cortex	SCA1 B05 transgenic mice	4 weeks, pre-symptomatic stage as well as 6–8-month old symptomatic mice	Fusion of MSCs with cerebellar neurons in the symptomatic mice only [120]
Human umbilical MSCs	Single bilaterial intracranial injection	Left and right cerebellar lobules IV	SCA1 B05 transgenic mice	4 weeks, pre-symptomatic stage	Improvement in molecular layer thickness, PC loss, and PC dendritic morphology; improvement in motor coordination [122]
MSCs	Single IT injection	Subarachnoid space	SCA1 knock-in mice (SCA1^154Q/2Q^)	5 weeks, pre-symptomatic stage	Larger axon and myelin sizes [21]
MSC conditioned medium	Single IT injection and/or multiple intravenous injections	Subarachnoid space and/or systemic circulation	SCA1 knock-in mice (SCA1^154Q/2Q^)	4 weeks, pre-symptomatic stage	Improvement in axon and myelin degeneration in the spinal neurons as well as improvement in motor coordination in all administration routes tested [123]
E13-E15 primordium mouse NPCs	Intracranial injection	Deep cerebellar nuclei	SCA1 B05 transgenic mice	6–9 weeks, early symptomatic stage	Improved motor performance and amelioration of PC dendritic morphology [125]
Mouse NPCs from subventricular zone	Single, bilateral intracranial injection	Cerebellar white matter	SCA1 B05 transgenic mice	5 or 13 weeks, pre-symptomatic; 24 weeks, symptomatic stage	Improvement in motor coordination, PC survival, PC function, and PC morphology in the symptomatic mice [124]

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
