# Peer review of "Therapeutic Strategies for Spinocerebellar Ataxia Type 1"

_biomolecules, 2023, doi:10.3390/biom13050788_

Round 1

Reviewer 1 Report

This comprehensive and well written manuscript offers a timely review, discussing the current therapeutic approaches for spinocerebellar ataxia type-1 (SCA1). The manuscript contains a very intricate description of the relevant literature, but I think it still needs some minor additions and improvements to meet its goals. I congratulate the authors for their commendable efforts.

1)     Authors should discuss the following study as it deals with regional vulnerability in SCA1 “Dual targeting of brain region-specific kinases potentiates neurological rescue in Spinocerebellar ataxia type 1”. This is important also in terms of why therapies need to be combinatorial.

2)     A characteristic dysfunction within the cerebellum which is important to mention and expand is the synaptic dysfunction associated climbing fiber-Purkinje neuron synapse and Parallel fiber-Purkinje synapse. Climbing fibers could also serve as a common framework to target all SCAs (Cleo J. L. M. Smeets, D.S. Verbeek, 2016), PMID: 28979190, Barnes J et al., 2011

3)     A better description of Purkinje neuron intrinsic excitability changes and how this could serve as a potential common pathway for targeting SCAs would be a good addition (https://www.sciencedirect.com/science/article/pii/S0306452220303778)

4)     A figure depicting cerebellar structure and the three cerebellar layers would help the review to reach non-specialists.

Reviewer 2 Report

The manuscript by Kerkhof et al. entitled “Therapeutic strategies for spinocerebellar ataxia type 1” is a timely review on preclinical and clinical studies for SCA1 therapy. This work critically reviews the genetic, pharmacological and cell replacement therapies tested for SCA1. It also clearly discusses the limitations for the development of a successful strategy and outlines future perspectives.

This manuscript is well written and organized; however, a few points require consideration. My comments are below:

1. On line 35, where is written “Disease causing alleles, have 39 or more CAG repeats with or without stabilizing CAT interruptions” should be corrected for “Disease causing alleles have 39 or more uninterrupted CAG repeats or large expansions with interruptions” (see GeneReviews, https://www.ncbi.nlm.nih.gov/books/NBK1184/).

 2. In Figure 2, the semicircle line seems to indicate the nucleus and that splicing is occurring in the cytoplasm.

 3. Figure 4, caption is repeated.

 4. In line 533, what mean the abbreviation ATXN2Q?

Reviewer 3 Report

This is an outstanding review of potential and present treatment options of SCA1. 
